# A Symmetry-Based Superposition Method for Planning and Surgical Outcome Assessment

**DOI:** 10.3390/bioengineering10030335

**Published:** 2023-03-06

**Authors:** Yu-Ching Hsiao, Jing-Jing Fang

**Affiliations:** Department of Mechanical Engineering, National Cheng Kung University, Tainan 701, Taiwan

**Keywords:** optimal symmetry plane, iterative closest point algorithm, superposition method, surgical outcome assessment method, patient-specific modeling

## Abstract

Computer-aided surgical planning has been widely used to increase the safety and predictability of surgery. The validation of the target of surgical planning to surgical outcomes on a patient-specific model is an important issue. The aim of this research was to develop a robust superposition method to assess the deviation of planning and outcome by using the symmetrical characteristic of the affected target. The optimal symmetry plane (OSP) of an object is usually used to evaluate the degree of symmetry of an object. We proposed a refined OSP-based contouring method to transfer a complex three-dimensional superposition operation into two dimensions. We compared the typical iterative closest point (ICP) algorithm with the refined OSP-based contouring method and examined the differences between them. The results using the OSP-based method were much better than the traditional method. As for processing time, the OSP-based contouring method was 11 times faster than the ICP method overall. The proposed method was not affected by the metallic artifacts from medical imaging or geometric changes due to surgical intervention. This technique can be applied for post-operative assessment, such as quantifying the differences between surgical targets and outcomes as well as performing long-term medical follow-up.

## 1. Introduction

Computer-assisted radiology and surgery have become important issues in clinical practice [1,2]. Computer-aided surgical planning has been widely used to enhance the safety and predictability of surgery and reduce operating time [3,4]. Using dedicated software, the affected site can be revealed by medical imaging and rendered as visible three-dimensional (3D) models on a computer. It is useful for the surgeon to observe the lesion before making a surgical plan, followed by fabricating patient-specific instruments to execute the plan and the surgical goal. However, most articles have focused on surgical planning before surgery [5,6], with fewer mentioning how to accurately validate the plan with surgical outcomes. Hence, patient-specific modeling was used to thoroughly investigate any deviations between surgical planning and outcomes to improve the use of patient-specific instruments to minimize the difference [7].

For the purpose of verifying and assessing the differences between surgical planning and outcomes, it is necessary to register the 3D models by superimposing medical imaging during planning and post-surgery [8]. Registration is also important and useful for post-surgery follow-up [9,10]. The most common approach has been the manual overlapping of corresponding anatomical features through observation [11]. This heavily relies on the reliability of experimental operators and their ability to conduct repeatable procedures; nevertheless, the labor-intensive and time-consuming nature of this method is a pitfall. The outcomes of the assessment may affect surgeons’ judgment in similar syndrome planning, resulting in changes in surgical outcomes.

With the progress of computational algorithms, automatic registration techniques between two subjects have improved rapidly. Proposed by Besl and McKay, the iterative closest point (ICP) method, which is the easiest and most popular technique to achieve the best match between two digital models [12], was first presented in 1992. The ICP method was used to minimize the differences between two point clouds for the purpose of co-registering two similar, but not necessarily identical, digital models. However, the results of this matching process may sometimes fail to converge to our expectations if the point clouds contain too many residuals or missing elements. Further improvement in the ICP method focused on removing inappropriate matching pairs [13] and increasing the speed of convergence [14].

The ICP algorithm has been frequently used in medical imaging registration, registration among more than two sources of medical imaging in computer-assisted surgery [15,16], and augmented reality in medical imaging projection on patient for exposed visualization during operations [17,18]. It has also been applied to simulate fracture reduction surgery for bone alignment animation [5,6,19]. The ridgelines of vertebral endplates were used to assess the progression of syndesmophytes by superimposing the associated vertebrae for long-term follow-up in patients with ankylosing spondylitis [20]. For pedicle screw insertion in spinal fusion operation, metal implants caused artifacts on medical imaging, which could lead to difficulties in quantifying deviations in screw trajectories between planning and real conditions. The ICP and its improved methods achieve perfect matching of two point clouds that have almost the same shape. However, for some deformed bones after surgery, the outcomes of kernel body superposition were not good enough using the ICP and its improved methods, and hence manual adjustment was subsequently conducted. This affected the assessment of surgical outcomes associated with planning. Even in vertebral implant insertion in spinal surgery, the affected vertebrae still retain a certain degree of symmetry. A superposition method integrated with the symmetric characteristic of the objects might be a solution for the above situations. Therefore, to create a suitable and unique representation of symmetry, determining the symmetry plane of the target is important.

Symmetry research on the spine has focused on evaluating the symmetry of each vertebra for scoliosis [21], measuring the axial rotation of the spine [22], and identifying the positions of vertebrae on an image using symmetric features [23]. However, most of the definitions in the symmetry plane are based on anatomical landmarks, which would vary from one person to another. Hence, this paper introduces a computerized voxel-based method to automatically locate the unique symmetry plane of an object, called the optimal symmetry plane (OSP). The symmetry ratio quantifies the degree of asymmetry of an object and determines the OSP, which is more reliable than the landmark-based and surface-based symmetry plane [24].

Superposition is useful for visualizing and quantifying the differences between two objects at different time points or the changes before and after an operation. The aim of this research was to develop a robust superposition method for the advanced comparison of two objects. 

## 2. Materials and Methods

### 2.1. Optimal Symmetry Plane

The degree of asymmetry is usually determined by comparing bilateral landmarks or contours. The symmetry plane is the most representative indicator. Based on computed tomography (CT), a voxel-based method was used in this study to find the OSP of bony tissues in the vertebrae. The OSP was computed using an optimization algorithm to find the plane that had the maximum number of voxel pairs of bilateral parts. Based on this unique OSP, the degree of asymmetry can be quantified. On one of the horizontal segmentation images of a cervical vertebra, for example, Figure 1a illustrates a possible symmetry line (in red) with its bilateral paired pixels indicated in red and its correspondence yellow crosses. Expanding the diagram to three dimensions by minimizing the voxel function,
(1)$$\mathrm{Min}f=1-\frac{∭ [v\left(x,\;y,\;z\right)\times \bar{v}\left(x,y,z\right)]dxdydz}{∭dxdydz},\;v\left(x,y,z\right)\times \bar{v}\left(x,y,z\right)\;=\left\{\begin{array}{c} 1,v\left(x,y,z\right)=\bar{v}\left(x,y,z\right)\\ 0,v\left(x,y,z\right)\neq \bar{v}\left(x,y,z\right) \end{array}\right.,$$
where 0≤f≤1, $v\left(x,y,z\right)$ is the original voxel located at $\left(x,y,z\right)$, and $\bar{v}\left(x,y,z\right)$ is its corresponding voxel in a given plane. The median plane of the CT gantry region is set to the initial guess plane of the OSP. The OSP is found by the plane that minimizes *f* [24]. 

The fourth cervical vertebra and its OSP in superior view are shown in Figure 1b.

### 2.2. OSP-Based Superposition Method

A 3D model was reconstructed using the marching cube method [25] to tessellate the faces of the model into a stereolithographic (STL) format. A highly dense point cloud was generated from the vertices of the tessellated model, which was used for superimposing in the next step.

The first step was to define the local coordinates of each object. Taking a vertebra as an example, as shown in Figure 2, the normal vector of the OSP of the vertebra represents the *x*-axis. In principal component analysis, the *z*-axis is the third component of the vertebra object projected to the OSP, and the *y*-axis is then obtained from the cross-product of the *x*- and *z*-axes [26].

The three points P_1_, P_2_, and P_3_ were defined on the OSP of the vertebrae for initial superposition, as shown in Figure 3. Assume that L_y_ is the length in the *y*-direction of the bounding box of the vertebral contour on the OSP. Based on the bending value method [27], a normalized bending value method was proposed for 3D application to obtain multiple turning points (blue dots in Figure 3) within the γ*L_y_ region from the y_min_ end:(2)$$B_{v}\left(i\right)=\max\left(\frac{\left|\left(V_{i+k}-V_{i}\right)+\left(V_{i-k}-V_{i}\right)\right|}{\left|V_{i+k}-V_{i-k}\right|}\right),$$
where $V_{i}$ indicates the sequential points on the contour, and *k* is half the width of the selected interval in $V_{i}$. The turning points were local maximum bending in a given interval of 2*k* and its bending value was larger than the given criteria of 1.8 × *k* in this case, and γ was set to 0.4. P_1_ is the geometric mean of the turning points, P_2_ is the centroid of the vertebra projected onto the OSP, and P_3_ is the maximum *z*-value vertex of the vertebra projected onto the OSP.

A transformation matrix was then generated via a translation to $P_{1}\left(x,y,z\right)$ and a rotation *T* formed by three unit vectors, giving
(3)$$T= [\begin{array}{cccc} u_{x} & v_{x} & w_{x} & 0\\ u_{y} & v_{y} & w_{y} & 0\\ u_{z} & v_{z} & w_{z} & 0\\ -x & -y & -z & 1 \end{array}],$$
where $\vec{u}\left(u_{x},u_{y},u_{z}\right),\;\vec{v}\left(v_{x},v_{y},v_{z}\right),$ $\vec{w}\left(w_{x},w_{y},w_{z}\right)$, $\vec{u}$, and $\vec{w}$ are the unit vectors of $\vec{P_{1}P_{2}}$ and $\vec{P_{1}P_{2}}\times \vec{P_{1}P_{3}}$, respectively. Suppose that $M_{A}$,$\;T_{A}$ and $M_{B},$ $T_{B}\;$ represent the vertebrae and their transformation matrices pre- and post-surgery, respectively. $M_{B}$ is transformed to $M_{A}$ by
(4)$$M=M_{B}\times T_{B}\times T_{A}{}^{-1},$$
as the initial superposition.

The vertices of the vertebra were projected onto its unique OSP to generate contour. Figure 4a,b show the contours of the pre- and post-surgical vertebrae projected onto their respective OSPs. The Moore neighbor-tracing algorithm [28] was used to sort the contour points. The pre-surgical vertebral contour A and the post-surgical vertebral contour B can be described as follows:(5)$$A=\left\{a_{1},\;a_{2},\cdots ,a_{i},\;\cdots ,a_{m-1},a_{m}\right\},$$
(6)$$B=\left\{b_{1},\;b_{2},\cdots ,b_{j},\;\cdots ,b_{n-1},b_{n}\right\},$$
where *m* and *n* denote the numbers of contour points in the pre- and post-surgical vertebrae, respectively.

The data structure of the two contours was built as a K-D tree [29]. The matching pair was found by searching the nearest point of contour *B* to a point on contour *A* through the K-D tree structure:(7)$$\min\left(d\left(a_{i},b_{j}\right)\right),a_{i}\in A;b_{j}\in B;i=1,\cdots ,m;\;j=1,\cdots ,n.$$

The K-D tree provided the nearest point searching. A rotation matrix *R* and a translation vector $\vec{t}$ exist such that the post-operative contour $b_{j}$ would transform to its matching pre-operative contour $a_{i}$, or in other words, $a_{i}=R\times b_{j}+\vec{t}$. However, the two superimposed contours are not exactly the same, which will cause deviations, expressed as follows:(8)$$E=\overset{m}{\underset{i=1}{\sum }}\left|R\times b_{j}+\vec{t}-a_{i}\right|$$

The deviation *E* is affected by both R and $\vec{t}$. To reduce the number of variables that need to be calculated, the centroids of the two contours can be moved to the coordinate origin. The translation vector is
(9)$$\vec{t}=\bar{a}-R\times \bar{b},$$
where $\bar{a}$ and $\bar{b}$ are the averages of the pre- and post-operative contours, respectively. Substituting $\vec{t}$ in Equation (9) into Equation (8) results in
(10)$$E=\overset{m}{\underset{i=1}{\sum }}\left|R\times \left(b_{j}-\bar{b}\right)-\left(a_{i}-\bar{a}\right)\right|.$$

Let $\bar{a_{𝜄}}=a_{i}-\bar{a}$, $\bar{b_{ȷ}}=b_{j}-\bar{b}$. The singular value decomposition (SVD) [30,31] was applied to calculate the rotation matrix *R* that minimizes the deviation *E*. Let the matrix
(11)$$H=\overset{n}{\underset{i=1}{\sum }}\bar{a_{𝜄}}{\bar{b_{ȷ}}}^{T},$$
where ${\bar{b_{ȷ}}}^{T}$ is the transpose of $\bar{b_{ȷ}}$. The SVD of matrix *H* is
(12)$$H=U\times \Lambda \times V^{T},$$
where U and V are mutually orthogonal matrices; Λ is a diagonal matrix with non-negative elements, and its diagonal values are the singular values of *H.* Using SVD, the rotation matrix that minimizes the deviation *E* can be obtained as follows:(13)$$R=V\times U^{T}.$$

Equation (13) is established when the determinant of *R* is 1. All the contour points are on the OSP and are coplanar. If the determinant of *R* is −1, *R* presents a reflection in the *z*-axis. The desired rotation matrix is
(14)$$R^{\prime }=V\times [\begin{array}{ccc} 1 & 0 & 0\\ 0 & 1 & 0\\ 0 & 0 & -1 \end{array}]\times U^{T},$$
which minimizes the deviation. By substituting into Equation (9), the translation vector $\vec{t}$ can be obtained. The above calculation performs one transformation of a matching pair of two contours. If the difference between the last two iterations is less than the threshold, the algorithm has converged. The criterion of convergence is that the difference between the overlapped areas of the two contours between two iterations is less than 0.001 mm^2^. In this study, we implemented the algorithms using C++ programming in a personal computer with a 3.6 GHz Intel Core i7-7700 CPU and 24 GB memory.

### 2.3. Clinical Evaluation

The clinical evaluations included a stability test for the proposed method and a comparison test of the traditional method and our proposed method. 

#### 2.3.1. Stability Test

The cervical vertebrae before and after laminoplasty were used to examine the robustness of locating the OSP in the stability test. During laminoplasty, the morphology of the target cervical vertebral lamina was changed. The OSPs of pre- and post-laminoplasty were generated first. Therefore, the vertebral body of the pre- and post-vertebrae were aligned to calculate the deviations between two OSPs in pre- and post-laminoplasty.

The inclusion criteria were adults who were diagnosed with cervical spinal stenosis and underwent cervical laminoplasty at Cathay General Hospital (CGH) in Taiwan. The patients who had previous operations on the affected site were excluded. All the patients had undergone CT both pre- and post-surgery. This was a retrospective study approved by the Institutional Review Board (IRB) at CGH in Taiwan, number CGH-P108137. Informed consent was obtained from all the subjects involved in this study. In total, 7 patients diagnosed with cervical myelopathy were recruited, 5 males and 2 females, aged from 31 to 67 yrs, with 23 affected cervical vertebrae in total. In terms of pre- and post-imaging sources, 10 vertebrae were from the same scanner, while the other 13 vertebrae were from different scanners.

The Mann–Whitney U test was used to compare the deviation angle and vertebral body deviation of the same and different CT groups. The type I error was α = 0.05, which assumed the null hypothesis H0 that the deviation angle and vertebral body deviation between the two groups were not different. If *p* < 0.05, it was considered a significant difference; otherwise, there was no significant difference between the same CT group and the different CT group.

#### 2.3.2. Comparison Test

We compared the proposed OSP-based superposition method and the traditional ICP algorithm. The test included patients who underwent posterior spinal fusion surgery and had metal implants in the affected sites. The metal implants caused severe artifacts on imaging that interfered with the superposition. Superimposing deviations, as well as the processing time in applying the ICP algorithm and methods, were obtained.

The inclusion criteria were adults who had been diagnosed with scoliosis or cervical spondylolisthesis and had undergone posterior pedicle screw fixation at National Cheng Kung University Hospital (NCKUH) in Taiwan and who also underwent pre- and post-surgical CT. The patients who had previous operations on the affected site were excluded. The trial was approved by the IRB in NCKUH, number B-ER-102-441. Informed consent was obtained from all the subjects involved in this study. Twenty-six patients comprising ten males and sixteen females were recruited. There were 55 vertebrae included in the comparison test in total.

The Wilcoxon signed-rank test was used to perform a paired comparison of the outcomes of superposition deviations and processing times of the two methods. The type I error was α = 0.05, which assumed the null hypothesis H0 that the superposition deviation and processing time between the two methods were not different. If *p* < 0.05, it was considered a significant difference; otherwise, there was no significant difference between the two superposition methods.

#### 2.3.3. Case Study

We used one of the vertebrae in the comparison test to demonstrate how the superposition results affect surgical precision verification. The items of the assessment included the entry point deviation distance (DD) in the horizontal (HDD) and sagittal views (SDD) and the deviation angle in 3D (DA) in the horizontal (HDA) and sagittal views (SDA).

## 3. Results

Table 1 shows the outcomes of the stability test. The average deviation angles of the two OSPs between the same and different CT groups were similar. The statistical results also showed that there was no significant difference between them. However, there was a significant difference between the two groups in superposition deviation between the two vertebral bodies. The average deviation angles of the two OSPs of affected vertebrae between pre- and post-surgery images was 0.45 ± 0.23°. The OSP method was robust and stable enough, even for imaging procedures using different brands of CT apparatus before and after surgery.

In the comparison test, 55 vertebrae were examined comprising 20 cervical, 24 thoracic, and 11 lumbar. Table 2 shows the superposition deviations and processing time of the two methods. Table 3 lists the statistical results of the comparison test. The vertebral deviation with the OSP-based method was better than that with the ICP method for the cervical (1.03 ± 1.46 vs. 1.14 ± 1.08 mm) and thoracic (0.89 ± 1.09 vs. 1.02 ± 1.00 mm) vertebrae. In addition, for the cervical and thoracic vertebrae, the vertebral body deviation with the OSP-based method was significantly better than with the ICP method (*p* < 0.05). Although the sample of lumbar vertebrae was small, it showed that the two methods had similar vertebral superposition results and vertebral body deviations, which showed no significant difference. Overall, for the 55 vertebrae, on average, the difference between the two methods for vertebral superposition and vertebral body deviation was about 0.09 mm and 0.26 mm, respectively. The processing time of the OSP-based and ICP methods were 2.96 and 34.88 s, respectively. Regarding the processing time, there was a significant difference for each group between the two methods. Thus, according to the statistical results, the processing time of applying the OSP-based contouring method was significantly faster than the traditional ICP algorithm. The OSP-based contouring method was 11 times faster than the ICP method on average.

Even if the results of the two superposition methods were similar numerically, the difference could be observed visually. Figure 5 shows the superposition results for the C2 vertebra before and after spinal surgery using ICP and OSP-based contouring methods. The yellow and blue vertebrae represent the pre- and post-surgical vertebra, respectively. Compared with the OSP-based method, the results of post-surgical vertebra superposition using the ICP method were biased in the +*y* direction. Figure 6 shows the superposition results of the C4 vertebrae in the laminoplasty case using ICP and OSP-based contouring methods. The significant deformation of the post-surgical vertebrae caused by plate insertion during operation could be observed. Due to the implant, the result of the ICP method was biased in the +*x* and −*z* directions, whereas the OSP-based contouring method converged to a better result visually.

Figure 7 shows the superposition comparisons of pedicle screw insertion in the planning (green dots) and post-surgical outcome (blue dots) of the left screw in a T9 vertebra. Table 4 lists the assessment results of surgical outcomes with the two methods in the T9 vertebra. In this case, the ICP method had a large superposition deviation in the z-direction, leading to a result biased in the +z direction. It can be observed from the sagittal view that the entry point position of the ICP method was higher than that of the OSP-based contouring method, which was reflected in the assessment that the SDD deviation was 4.84 mm.

## 4. Discussion

This study consisted of three parts: the definition of the optimal symmetry plane; the descriptions of the OSP-based contouring method; and the clinical evaluation including a stability test, a comparison test, and a case study. In the stability test, we retrospectively analyzed twenty-three laminoplasty cases to evaluate the robustness of the OSP. The average deviation of the closest points before superimposing the pre- and post-surgical vertebral bodies for the same and different CT groups were 0.19 ± 0.03 mm and 0.52 ± 0.22 mm, respectively, which showed a significant difference. The shape of the affected part as reconstructed using different CT imaging machines would be slightly different. The average angular deviations between pre- and post-surgical OSPs for the same and different CT groups were 0.43 ± 0.26° and 0.45 ± 0.20°, respectively, which were almost the same, and statistical analysis showed no significant difference. This result substantiates that the OSP provides a robust and unique method for representing the symmetry plane of an object.

For superposition, the initial position of the superposition process greatly impacted the iteration time and even the results. The initial superposition process of the proposed method successfully brought the two objects closer, in some cases close to the final position. Unlike the improved ICP algorithm [13], the proposed method used the OSP to find the closest features, i.e., the centralized contours of the two superimposed objects. Therefore, the complex 3D computations were converted into 2D to reduce time complexity and increase performance. According to the comparison test, the processing time of the proposed method was merely 3 s on average, which is significantly faster than the ICP algorithm due to reducing 3D into 2D calculations.

For the comparison test involving pedicle screw insertion in spine surgery, the deviations of the superposition between ICP and the proposed method were not much different, but only 29% of cases converged to the satisfied position with the ICP method. However, all of the fifty-five vertebrae successfully converged to the satisfied position with the maximum overlapping contour area using the OSP-based contouring method. This is because when the image quality of the CT scan was poor or when the metal implant artifacts greatly impacted the superposition processes, the ICP algorithm converged to a local but not a global minimum position. The deviations of superposition were numerically similar due to the deviation on average, whereas deviations between the two vertebrae were obviously observed, as shown in Figure 5 and Figure 6.

According to the statistical results, non-significant differences occurred in both vertebrae and vertebral body deviations in the lumbar group. The lumbar spine is larger than the other vertebral levels. After screw implant surgery, artifacts are relatively small compared with the lumbar vertebrae. Hence, the superposition result of using the ICP method was not significantly different from the proposed method. For the cervical or thoracic vertebrae, the artifacts contained a certain volume ratio of the vertebra, which may result in convergence to the local minimum position using the ICP algorithm. We can conclude that the ICP algorithm is suitable for superimposing two almost identical objects. However, for superimposing two slightly deformed objects, the proposed OSP-based contouring method might be the appropriate choice. The limitation of the proposed method is that when the two objects have global deformations resulting in significant changes in the contours, superposition may fail. Additionally, if the deformation changes the original symmetric features, the OSP of the two objects might deflect and result in failed superposition.

Further application of the superposition method enables researchers to evaluate accuracy in post-surgical outcomes. For example, when superimposing the pre- and post-surgical vertebrae for laminoplasty, the variation in the circumference, cross-sectional area, and volume of the vertebral canal after operative expansion can be quantified. Superimposing the pre- and post-surgical vertebrae with a vertebral compression fracture enables the calculation of the vertebral height variation to evaluate surgical outcomes for longitudinal follow-up. Another application is to superimpose the pre-operative planning to the post-surgical vertebrae for pedicle screw insertion to evaluate surgical precision. In addition, the proposed method is applicable not only for superimposing pre- and post-operative sites but also for superimposing sites representing the long-term follow-up of non-surgical treatments. For those patients who have undergone needless surgery for ankylosing spondylitis, syndesmophyte growth can be observed by superimposing multiple vertebrae.

## 5. Conclusions

In this research, we developed a robust superposition method for an advanced comparison of two objects. The OSP-based contouring method was developed to automatically superimpose two objects with partial deformations. The outcomes of the proposed method were not affected by the metallic artifacts from implants or geometric changes due to operation. The stability test retrospectively reviewed laminoplasty cases to examine the uniqueness and robustness of the OSP. The OSP is a robust and unique method for representing the symmetry plane of an object. Cases of pedicle screw fixation in spine surgery were used to compare the deviation and efficiency of the ICP and OSP-based contouring methods. Compared with the ICP method, the superposition deviation of the OSP-based contouring method was better, and its processing time was approximately 11 times faster. The proposed method can be applied for post-operative assessment, such as for the quantification of the deviation between targets and surgical outcomes and the examination of the development of affected sites between two time points in a long-term follow-up.

## Figures and Tables

**Figure 1 bioengineering-10-00335-f001:**
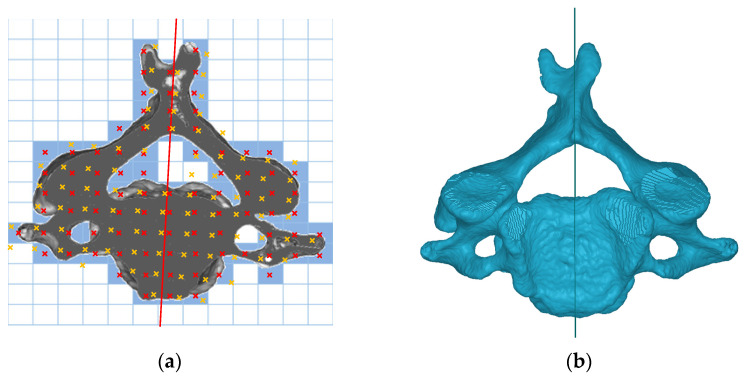
(**a**) Possible symmetry line on a cross-sectional image; (**b**) superior view of a C4 vertebra and its OSP.

**Figure 2 bioengineering-10-00335-f002:**
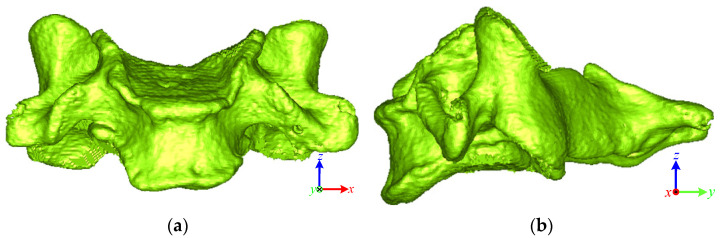
Local coordinate of the vertebra: (**a**) anterior view; (**b**) lateral view.

**Figure 3 bioengineering-10-00335-f003:**
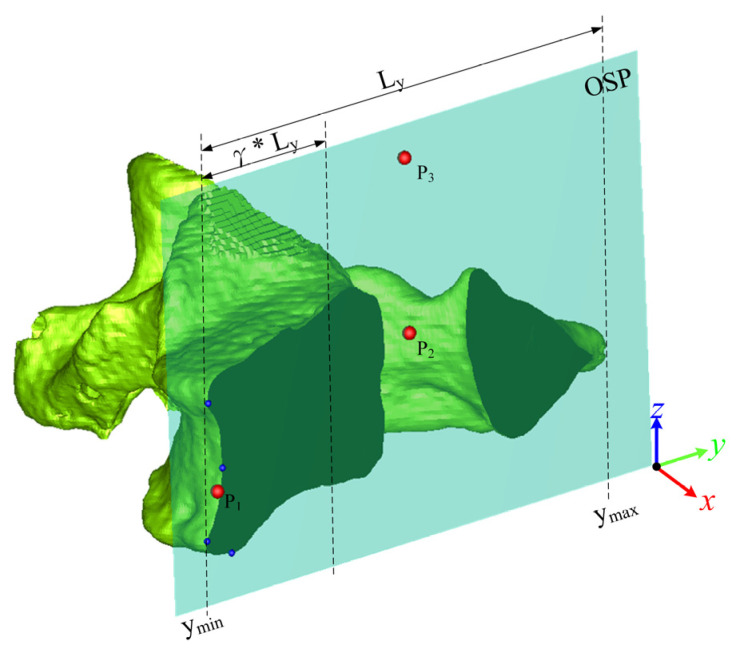
The three vertices on the OSP.

**Figure 4 bioengineering-10-00335-f004:**
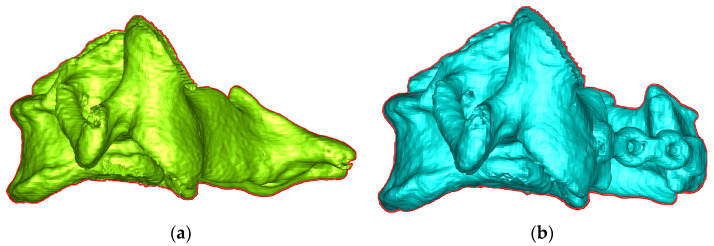
Lateral view and contour of vertebra in (**a**) pre-OP and (**b**) post-OP.

**Figure 5 bioengineering-10-00335-f005:**
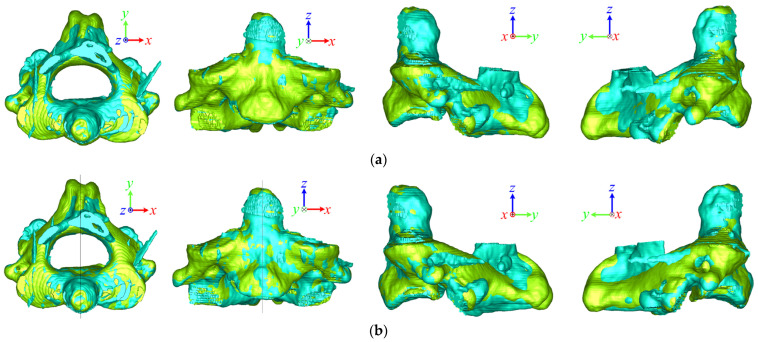
Superposition comparison of the (**a**) ICP and (**b**) OSP-based contouring methods for a pre- and post-surgical C2 vertebra in the pedicle screw fixation case.

**Figure 6 bioengineering-10-00335-f006:**
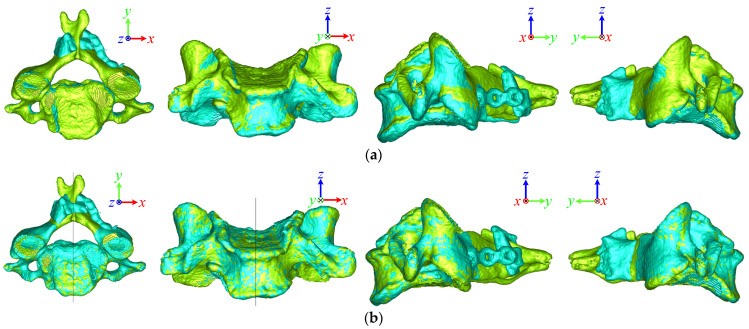
Superposition comparison of the (**a**) ICP and (**b**) OSP-based contouring methods for a pre- and post-surgical C4 vertebra in the cervical laminoplasty case.

**Figure 7 bioengineering-10-00335-f007:**
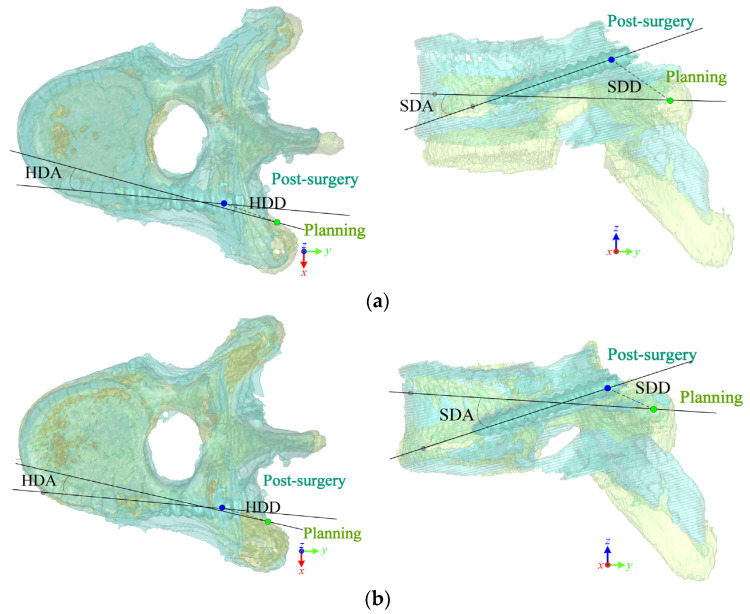
Comparison of planning and post-surgical screw trajectories at left screw in the T9 vertebra after superimposing via (**a**) ICP and (**b**) OSP-based contouring methods.

**Table 1 bioengineering-10-00335-t001:** Comparisons of deviation angles and superposition deviations between two groups.

	Deviation Angle of Two OSPs	Superposition Deviation Btw Two Vertebral Bodies
Same CT group	0.43 ± 0.26°	0.19 ± 0.03 mm
Different CT group	0.45 ± 0.20°	0.52 ± 0.22 mm
Statistical analysis (*p*-value)	0.642	<0.001 *
23 vertebrae	0.45 ± 0.23°	0.37 ± 0.24 mm

* *p* < 0.05.

**Table 2 bioengineering-10-00335-t002:** Performance and deviation comparisons of the ICP and OSP-based contouring methods.

	ICP Algorithm			OSP-Based Contouring Method		
	Vertebrae Deviation (mm)	Vertebral Body Deviation (mm)	Processing Time (s)	Vertebrae Deviation (mm)	Vertebral Body Deviation (mm)	Processing Time (s)
Cervical	1.14 ± 1.08	0.97 ± 0.80	25.07 ± 22.68	1.03 ± 1.46	0.58 ± 0.61	2.92 ± 1.82
Thoracic	1.02 ± 1.00	0.90 ± 0.74	36.99 ± 19.40	0.89 ± 1.09	0.63 ± 0.57	2.99 ± 1.48
Lumbar	0.98 ± 1.07	0.83 ± 0.75	48.13 ± 23.81	1.00 ± 1.19	0.81 ± 0.76	2.92 ± 1.84
55 vertebraeon average	1.05 ± 1.04	0.91 ± 0.76	34.88 ± 23.18	0.96 ± 1.25	0.65 ± 0.62	2.96 ± 1.67

**Table 3 bioengineering-10-00335-t003:** Wilcoxon signed-rank test results of pairwise comparisons between the ICP and OSP-based contouring methods.

	Vertebral Deviation	Vertebral Body Deviation	Processing Time
Cervical	0.040 *	0.001 *	<0.001 *
Thoracic	0.061	<0.001 *	<0.001 *
Lumbar	0.283	0.397	0.003 *
55 vertebrae	0.020 *	<0.001 *	<0.001 *

* *p* < 0.05.

**Table 4 bioengineering-10-00335-t004:** How the superposition results affect surgical outcome assessments in the T9 vertebra.

Assessments	ICP Method	OSP-Based Contouring Method	Difference
DD	7.44 mm	2.50 mm	4.94 mm
HDD	3.65 mm	2.19 mm	1.46 mm
SDD	6.53 mm	1.69 mm	4.84 mm
DA	27.82°	30.25°	−2.43°
HDA	3.74°	4.56°	−0.82°
SDA	27.90°	30.22°	−2.32°
Process time	36 s	1.10 s	35 s

DD: the 3D deviation distance of the entry point; HDD: horizontal deviation distance between two entry points; SDD: sagittal deviation distance between two entry points; DA: 3D deviation angle between two trajectories; HDA: horizontal deviation angle between two trajectories; SDA: sagittal deviation angle between two trajectories.

## Data Availability

Not applicable.

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
