# Peer review of "A Symmetry-Based Superposition Method for Planning and Surgical Outcome Assessment"

_bioengineering, 2023, doi:10.3390/bioengineering10030335_

Round 1

Reviewer 1 Report

This manuscript reported a method for symmetry based superposition, and then assessed this method via a stability study and and a comparison study. The topic is interesting. The studies demonstrated merits. But,  the contents are organized poorly.

Introduction is lengthy and repetitive.  The objective was stated but the hypothesis has not been clearly defined.

L134-L151 reported a study to test the robustness of the OSP method. Is it the stability study as mentioned in abstract? Why its results inluded in the method section? One-sample t-test was used to "determine the sample size" does not make sense.

In the description of the symmetry-based superposition method,  glad to read these mathematic formula, but please clearly define each concept, terminology,  and be consistent.

The comparison study did not described in method at all, but reported in results section instead. Why so? To evaluate the effects of two different methods, please consider to perform an equivalence study statistically. Simply report group means and standard deviation may not suffice to prove whether these two methods are equivalent or not.  

Fig.7  and table 2 were introduced in discussion , NOT in method and results. Where are they come from? Which study? Also, why not report standard devision in Table 2?

In all, I do not think this manuscript is ready for publication. Although this method and studies have merits, extensive organizing,  editing, and proofreading are needed.

Reviewer 2 Report

This manuscript is a well written description and pictorial essay using OSP-based contouring method for surgical outcomes.

I have some issues that should be considered by the authors to make their work more robust.

Material and Methods

Where did the images come from? Patient consent?

Give details regarding the monitor and computer used in the analysis. Who performed the image processing

Discussion:

The Discussion is slightly disorganized, and it is difficult to determine what points are based on the author’s study, their conjecture, or previously published literature.

The Limitations section should be expanded to include concerns raised in Weaknesses.

Reviewer 3 Report

The authors propose a robust superposition method to assess the deviation of planning and outcome by using the symmetrical characteristic of the affected target (in this case, bones). They consider that most of the analyzed targets are simmetrical in majority, and the alignment is calculated based on the optimal symmetry plane (OSP). This was a very clever idea and the results are pretty impressive. The paper is well writen with minor writing errors listed as follows.

"Aim of this" -> "The aim of this"

"method did not been affected" -> "method had not been affected"

"a reliable analyses" -> "a reliable analysis"

"respectively." -> ", respectively."

"vertebrae respectively." -> "vertebrae, respectively."

"mm respectively." -> "mm, respectively."

"mm respectively." -> "mm, respectively."

"convert a complex 3D computations" -> "convert complex 3D computations"

"contours, of the" -> "contours of the"

In order to improve the clarity of the paper for readers, some points need to be addressed.

What was the hardware infrastructure used for processing the evaluations? Please describe the configuration of the computer used to acquire the processing times from Table 1.

What is the format of the input files used in the work? Is it a common .pcd (point cloud data)? Authors say the algorithm is based on voxels, but do not provide more details or even link to the data used (if possible) so that readers could reproduce the work.

Round 2

Reviewer 1 Report

Authors made considerable revision and solved major structure issues in v2. This allows me to conduct further review. My view now is that this manuscript reported a strong study and has merits to be published.

The major issue, however, is its clarity and repeatability, which need to be further addressed. I provided detailed and extensive comments, questions, and suggestions  in a PDF file for your consideration. However, my review should not be considered as limits to your further revision, which should follow the course of this study itself. You are encouraged to make any necessary modifications and revisions. I am glad to re-review again. 

Reviewer 2 Report

I have reviewed the re-submission and the authors have carefully amended their manuscript following my suggestions.

Author Response

The authors appreciate the suggestions for this manuscript from the reviewer.

Round 3

Reviewer 1 Report

V3 is better than v2.  We are getting there.

V3 should be a clean version. 

Please consider the following:

- Both " any" and " all" artifacts are too strong. What you dealt with  are two: metallic artifacts from Implants, and surgical movement. This manuscript did not deal with motion artifacts, for instance. Please do not use " any", "all". I suggest be specific.  

- Formula (1)  and its introduction is incomplete.  A simple formula does not mean it is a optimization. I can not repeat this without further information. Please provide further information to ensure reader can repeat what you did.

- Please use "time point".  Not " time zone", not " time stage", Not " period of time".   

- Why not include "comparison study"  in the caption of both Table 2 and table 3?  Also, the captions of figures and tables should be self-contained, that is , "complete, or having all that is needed, in itself."

-  Please rewrite the Conclusion,so as to be consistent with abstract and study aim in the last sentence of the introduction.  Please try to not over interpret this method without testing.

In my view, this is an automaticable but complex approach. Its application needs to be further investigated prior to make further judgment. It has its merits to be published, though.
